# MAR-YOLOv9: A multi-dataset object detection method for agricultural fields based on YOLOv9

**Dunlu Lu**[1]☯, **Yangxu Wang**[1,2]☯*

**1** College of Robotics, Guangdong Polytechnic of Science and Technology, Zhuhai, Guangdong, China,
**2** Department of Network Technology, Guangzhou Institute of Software Engineering, Conghua, Guangdong, China

☯ These authors contributed equally to this work.
* wangyx6432@gmail.com

**Data Availability Statement:** The curated data and code can be accessed at the following link: https://github.com/YangxuWangamI/MAR-YOLOv9.

**Funding:** This work was supported by the 2022 Guangdong Province Ordinary Universities

## Abstract

With the development of deep learning technology, object detection has been widely applied in various fields. However, in cross-dataset object detection, conventional deep learning models often face performance degradation issues. This is particularly true in the agricultural field, where there is a multitude of crop types and a complex and variable environment. Existing technologies still face performance bottlenecks when dealing with diverse scenarios. To address these issues, this study proposes a lightweight, cross-dataset enhanced object detection method for the agricultural domain based on YOLOv9, named Multi-Adapt Recognition-YOLOv9 (MAR-YOLOv9). The traditional 32x downsampling Backbone network has been optimized, and a 16x downsampling Backbone network has been innovatively designed. A more streamlined and lightweight Main Neck structure has been introduced, along with innovative methods for feature extraction, up-sampling, and Concat connection. The hybrid connection strategy allows the model to flexibly utilize features from different levels. This solves the issues of increased training time and redundant weights caused by the detection neck and auxiliary branch structures in traditional YOLOv9, enabling MAR-YOLOv9 to maintain high performance while reducing the model's computational complexity and improving detection speed, making it more suitable for real-time detection tasks. In comparative experiments on four plant datasets, MAR-YOLOv9 improved the mAP@0.5 accuracy by 39.18% compared to seven mainstream object detection algorithms, and by 1.28% compared to the YOLOv9 model. At the same time, the model size was reduced by 9.3%, and the number of model layers was decreased, reducing computational costs and storage requirements. Additionally, MAR-YOLOv9 demonstrated significant advantages in detecting complex agricultural images, providing an efficient, lightweight, and adaptable solution for object detection tasks in the agricultural field. The curated data and code can be accessed at the following link: https://github.com/YangxuWangamI/MAR-YOLOv9.

Characteristic Innovation Project (Grant no. 2022KTSCX251) and the open fund projects of Hubei Key Laboratory of Intelligent Robot (Wuhan Institute of Technology) (Grant no. HBIR 202209).

**Competing interests:** The authors have declared that no competing interests exist.

## Introduction

The rapid development of deep learning technology [1] has led to significant progress in object detection across multiple domains [2–4]. Modern object detection methods based on Convolutional Neural Networks (CNNs) have achieved many important technical breakthroughs through their powerful feature extraction capabilities. However, existing technologies still face performance bottlenecks when dealing with diverse scenarios, especially in diverse application scenarios such as different image styles, lighting conditions, image quality, and camera perspectives.

The reasons for the performance decline can be summarized as follows: Diversity of image styles: Images in different application scenarios may have completely different visual features, such as changes in shape, texture, and color. Uncertainty of lighting and environmental conditions: Changes in lighting, shadows, and reflections affect the quality of images and the performance of object detection. Variability of image acquisition conditions: Different camera perspectives and image quality lead to diversity in image content. Variations in target size and posture: Different shooting angles may cause significant changes in the appearance and scale of targets, increasing the complexity of the detection task. In response to these challenges, the motivation of this study is to develop an object detection method that can adapt to the specific needs of the agricultural field. The agricultural field is characterized by a wide variety of crops, significant changes in appearance during the growth cycle, and a complex and variable farmland environment. These factors together contribute to the complexity of the object detection task. Moreover, agricultural applications require the detection model to be lightweight and real-time for deployment on resource-constrained agricultural equipment.

To address these challenges, this study proposes a lightweight, domain-adaptive enhancement object detection method based on YOLOv9 [5], named Multi-Adapt Recognition-YOLOv9 (MAR-YOLOv9), to meet the characteristics and needs of the agricultural field. We have improved the model structure, optimized the algorithm process, and adjusted the training strategy to better adapt MAR-YOLOv9 to the characteristics and needs of the agricultural field, enhancing the model's generalization and robustness. Specifically, we have optimized the traditional 32x downsampling Backbone network and innovatively designed a 16x downsampling Backbone network. This improvement not only preserves the necessary feature information but also reduces computational complexity and improves the efficiency of feature extraction. Secondly, we introduced a more streamlined and lightweight Main Neck structure, further reducing the model's computational cost by simplifying the structure and optimizing parameters. At the same time, we innovated in feature extraction, up-sampling, and Concat connection methods, making the model more efficient in feature fusion and information transfer, allowing MAR-YOLOv9 to better adapt to the complex and variable environment of the agricultural field. To comprehensively evaluate the performance of the MAR-YOLOv9 model, we selected four publicly available datasets with different characteristics as benchmarks for experimental verification. These datasets vary in scale and density and have different resolutions, clarity, and lighting conditions, which can more objectively evaluate the model's domain adaptability. We conducted comparative experiments with other advanced methods, especially with the benchmark model YOLOv9, for detailed training and evaluation. The experimental results show that the proposed MAR-YOLOv9 performs excellently in terms of performance and efficiency, surpassing other computer vision methods while maintaining sufficient generality.

The main contributions of this paper are as follows:

1. Innovatively designed a 16x downsampling Backbone network: Solved the increased training time and redundant weights caused by the detection neck and auxiliary branch structures in YOLOv9.

2. Optimized the design of the main detection neck: Reduced the model's computational burden and the number of parameters, making the model more suitable for resource-constrained environments while maintaining high detection accuracy on multiple plant datasets.

3. Proposed the MAR-YOLOv9 model with agricultural domain adaptability: Compared with other procedural methods on multiple plant detection and counting datasets, verifying its performance and adaptability.

This section (Introduction) introduces the research background and motivation of this paper. The Related Work section reviews the work and highlights issues. The Model and Methods section provides a detailed description of the model design. The Materials and Experiments section describes the data sets used, provides details of the experiments, and gives the results. The Discussion section discusses the findings. The Conclusion section summarizes the whole paper and puts forward the future research direction.

## Related work

In the field of deep learning object detection, there are mainly two types of methods. The first category includes two-stage object detection algorithms, such as Spatial Pyramid Pooling Networks (SPP-Net) [6], Regions with Convolutional Neural Networks (R-CNN) [7], and Faster R-CNN [8]. These algorithms typically have higher detection accuracy but are slower in detection speed due to their two-stage nature. In contrast, the second category is single-stage object detection algorithms, such as You Only Look Once (YOLO) and CenterNet [9], which offer faster detection speeds, albeit with a slight sacrifice in accuracy. Due to the high demand for detection speed in most tasks, single-stage algorithms have more advantages in practical applications.

Facing the performance challenges of deep learning models in object detection tasks, researchers have adopted various strategies to enhance the model's generalization and accuracy. For example, Zhou et al. [10] proposed a novel method for salient object detection using multi-visual perception, reflecting the rapid recognition of the human visual system and focusing on impressive objects/regions in complex scenes. By fusing multi-visual perception features and leveraging superpixel segmentation, the detection performance in complex scenes was effectively improved. Liu et al. [11] introduced an innovative Adaptive Multi-Scale Feature Enhancement and Fusion Module (ASEM) algorithm, which enhanced the object detection performance of remote sensing images through fine multi-scale feature fusion, achieving mAP improvements of 74.21% on the DOTA-v1.0 dataset and 84.90% on the HRSC2016 dataset. Yue et al. [12] proposed a lightweight detection network for single-class multi-deformable targets, YOLO-SM, which significantly improved the accuracy and generalization capabilities of object detection while maintaining high speed, through the DCM module and GMF feature fusion structure. Kim et al. [13] proposed a new anchor-free object detection method focused on detecting minor defects on the surface of apples. Additionally, Wang et al. [14] proposed a dual-branch structure to enhance the handling of cross-domain classification problems by combining Graph Convolutional Networks (GCN) and Convolutional Neural Networks (CNN), with one branch using GCN to capture global information and the other using CNN to focus on local features. This dual-branch structure enhances the model's ability to handle cross-domain classification problems, combining a dynamic weighted hierarchical loss

function to address challenges in cross-domain classification. L et al. [15] improved the YOLOv5 algorithm by proposing RSI-YOLO, which enhances model performance by introducing an attention mechanism and a bidirectional feature pyramid network, as well as an EIoU loss function. Similarly, Han et al. [16] designed an end-to-end attention mechanism specifically for crop mapping in time-series SAR imagery. Although these methods have made progress in cross-domain issues, they still have limitations such as dependence on labeled data, high computational resource requirements, and a lack of universality. Future research needs to focus on addressing these challenges to further enhance the performance and applicability of object detection in the agricultural field.

In the agricultural field, remote sensing image object detection tasks have their uniqueness. In recent years, researchers have proposed a series of innovative methods in response to the particularities of the agricultural field. For example, Zhou et al. [17] studied the issue of remote sensing image registration, and there are studies on efficient detection and counting of corn ears [18, 19] and wheat ears [20, 21], which not only demonstrate the broad application prospects of object detection in the agricultural field but also provide strong support for solving the special challenges of the agricultural field. Due to the complexity of the agricultural environment, object detection models need to have strong feature extraction and classification capabilities. Firstly, there is a wide variety of agricultural crops, and the feature differences between different crops are significant, so the object detection model needs to have stronger feature extraction and classification capabilities. Secondly, the agricultural environment is complex and variable, and factors such as lighting conditions, occlusions, and background interference can affect the performance of object detection. In addition, considering the actual needs of agricultural production, object detection models also need to have the characteristics of lightweight and real-time to run on resource-limited agricultural equipment. In response to these challenges, we need to find an object detection method that is both accurate and real-time. Fortunately, YOLOv9 [5], as the latest member of the YOLO series, is a good choice. It has further improved the accuracy and real-time performance of object detection by improving network structure and algorithm process on the basis of inheriting the advantages of previous models, achieving significant results in general object detection tasks. However, when YOLOv9 is directly applied to the agricultural field, it still encounters some special challenges, such as crop diversity and environmental complexity. Therefore, it is necessary to further optimize and adapt existing object detection methods to meet the needs of the agricultural field and improve their performance and applicability in agricultural scenarios.

## Model and methods

In this section, we will elaborate on the design principles, structural features, and optimization methods of the MAR-YOLOv9 model.

### Model construction

Agricultural object detection faces challenges of crop diversity and complex environments. To address these challenges, we propose the MAR-YOLOv9 model, which is based on the deep learning Backbone-Neck structure and employs a dual-path detection structure [5] to reduce information loss. The model consists of two parts: the Backbone and the Neck, as shown in Fig 1. The Neck part includes primary and auxiliary detection branches. The key configurations of the model will be introduced next.

**Backbone.** The Backbone part is responsible for extracting features from the input image. In this model, the Backbone uses a lightweight Convolutional Neural Network (CNN) structure, which progressively extracts multi-scale features from the image by stacking multiple

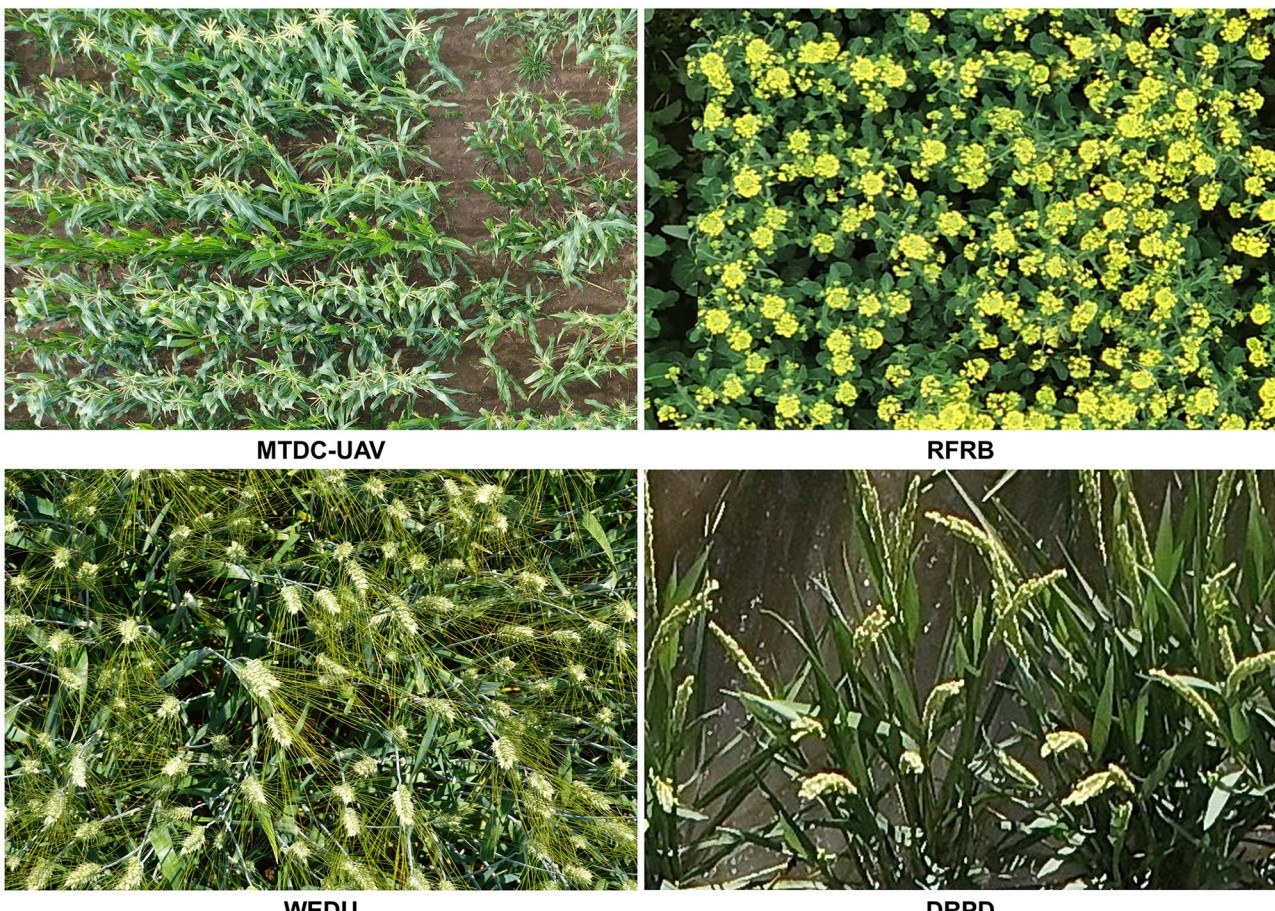

**Fig 1. Architecture of MAR-YOLOv9.**

convolutional layers, activation functions, and downsampling layers. The specific configuration is as follows: The Silence module [5] is located at the very front of the network structure and is responsible for receiving raw image data and performing preprocessing to prepare for subsequent feature extraction. It does not perform any substantial operations, aiming to preserve the feature information of the original image so that the primary and auxiliary detection necks can fully utilize this information for object detection, greatly facilitating the invocation of dual-trunk structures and other tasks. Subsequently, the model uses multiple Conv layers to perform convolution operations. Each Conv layer uses a 3 × 3 convolutional kernel to extract features from the image and reduces the computational load and dimensions of the feature map through downsampling operations with a stride of 2. The process of the convolution operation can be represented by the following formula (1):

$$X_{\text{out}} = \sigma\left(\sum_{i=1}^{N}(X_{\text{in}} \cdot W_i + b_i)\right), \tag{1}$$

where $\sigma$ represents the activation function, * denotes the convolution operation, $W$ is the convolutional kernel, $b$ is the bias term, and $X_{out}$ is the output feature map. When the input image size is $H_{in} \times W_{in}$, the convolutional kernel size is F × F, the padding is $P$, and the stride is $S$, the

size of the output feature map can be calculated using the following formulas (2) and (3):

$$H_{out} = \left\lfloor \frac{H_{in} + 2P - F}{S} \right\rfloor + 1, \tag{2}$$

$$W_{out} = \left\lfloor \frac{W_{in} + 2P - F}{S} \right\rfloor + 1, \tag{3}$$

This design helps to reduce the computational complexity of the model while retaining sufficient feature information to support subsequent detection tasks. To fully utilize feature information at different levels, feature fusion layers are introduced at different stages of the Backbone. These fusion layers merge features from different levels, allowing the model to focus on both low-level detail information and high-level semantic information. This enables the model to better adapt to the detection needs of targets at different scales and improve overall detection performance.

After the configuration of the Backbone, the model includes 4 downsampling convolutional layers and 3 RepNCSPELAN4 feature extraction layers. Through the stacking and combination of these layers, the input image will complete a 16x downsampling, and the feature map is reduced to 1/16 of the original image size, outputting multiple feature maps at different depths to provide rich feature information for subsequent detection tasks.

**Neck.** The Neck network part is responsible for further processing the deep features extracted by the Backbone to convert them into accurate detection results. The Neck network employs multi-scale feature fusion technology, combined with the design of a reversible auxiliary branch, effectively enhancing the model's detection capability for targets at different scales. The specific configuration is as follows:

Firstly, in the primary detection neck, we use an innovative 16x downsampled feature map, which first undergoes the SPPELAN module [5]. By expanding the receptive field, it enhances the abstraction and expression capabilities of the features. Subsequently, through the feature upsampling operation Upsample, the size of the feature map is enlarged using nearest-neighbor interpolation to facilitate fusion with larger feature maps. This fusion operation is implemented by the Concat module, which effectively integrates feature information at different levels, thereby improving the model's detection capability for multi-scale targets. After further processing by the RepNCSPELAN4 feature extraction layer, the feature map is sent to the Detect detection head for the final object detection.

In addition to the primary detection neck, we also employ an auxiliary detection neck to address the issue of information loss caused by the increased depth of the network. The introduced reversible auxiliary branch, through the CBLinear and CBFuse modules [5], where CBLinear processes the input data through a convolutional layer and then splits the results into multiple tensors according to a specified list of output channels, and CBFuse accepts a series of feature maps and an index list, adjusts the size of these feature maps to the same size as the last feature map using nearest-neighbor interpolation, and then sums them along the batch dimension to achieve feature fusion. When used in combination, they implement multi-path feature transmission, effectively solving the problem of information loss in deep networks and enhancing the detection accuracy for small targets. Specifically, the CBLinear module feature extractor extracts features from the 4x, 8x, and 16x downsampled layers of the Backbone. Subsequently, the CBFuse module unifies the size of the feature maps to the same scale through upsampling operations, ensuring that they can be fused in the same spatial dimension, followed by an addition operation to achieve the fusion of multi-level auxiliary information.

This process can be represented by the following formula (4):

$$X_{outfeatures} = CBLinear(X_4,\ X_8,\ X_{16}),\tag{4}$$

where $X_4$, $X_8$, and $X_{16}$ are the feature maps obtained through 4x, 8x, and 16x downsampling, respectively, and $X_{outfeatures}$ is the auxiliary feature extracted by the CBLinear module. Finally, the MAR-YOLOv9 model adopts a multi-scale detection strategy, passing five feature maps of different scales into the detection head, which generates target bounding boxes and class information corresponding to the scale of the feature maps, thereby outputting detection results of different scales and better adapting to targets of different sizes. Moreover, the end-to-end training strategy simplifies the training process and improves the model's detection efficiency by directly predicting the target's bounding boxes and classes from the fused feature maps through the Detect detection head. Fig 2 shows the structure of the main modules RepNCSPE-LAN4, CBLinear, and Adown in MAR-YOLOv9, providing a more intuitive understanding of the design philosophy and implementation methods of each module in the MAR-YOLOv9 model. The optimization and innovative design of these modules collectively enhance the model's detection performance, making MAR-YOLOv9 outstanding in plant object detection and counting tasks.

In summary, the MAR-YOLOv9 model adopts a 16x downsampled Backbone design, extracts features in dual-path detection, and through innovation in the connection methods of model modules, not only introduces a multi-scale feature fusion strategy into the decoding layer to ensure that the model can capture and integrate key information from different depths but also designs a reversible auxiliary branch and feature map fusion module to enhance the flow of information and the richness of feature expression, giving MAR-YOLOv9 an advantage in plant object detection and counting tasks.

**Activation functions.** In deep learning, activation functions play a crucial role in the performance and convergence speed of the model, determining whether the network can learn non-linear relationships. Common activation functions include Sigmoid Linear Unit (SiLU) [22], Rectified Linear Unit (ReLU) [23], Leaky ReLU [24], and hyperbolic tangent (Tanh), etc. Each activation function has its unique mathematical form and characteristics. The SiLU activation function combines linear and non-linear properties, and its definition is given by the

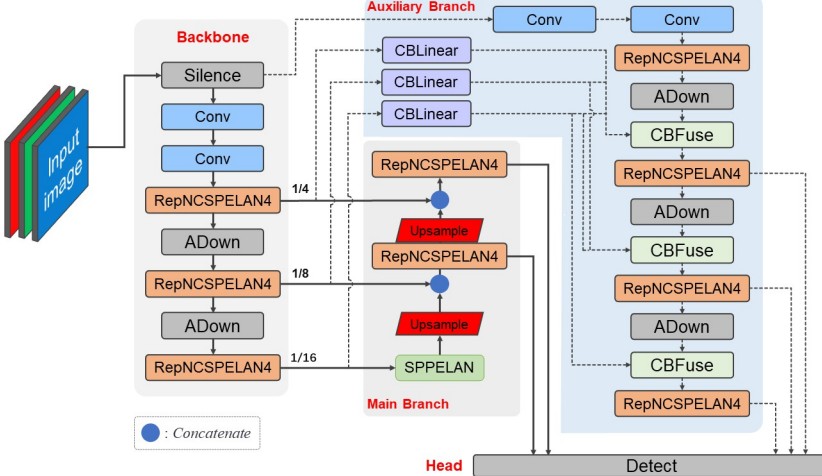

**Fig 2. Main module structure.**

following formula (5):

$$SiLU(x) = x \cdot \sigma(x), \tag{5}$$

where $\sigma(x)$ is the sigmoid function: $sigmoid(x) = \frac{1}{1+e^{-x}}$.

SiLU dynamically adjusts the scaling of the input $x$ through the output of the sigmoid function, preserving the linear part of the input information while introducing non-linear factors. This design helps to avoid the problem of gradient disappearance and, to some extent, prevents the "death" of neurons.

When choosing an activation function, we need to weigh based on the specific requirements of the task and the characteristics of the model. For example, in the MAR-YOLOv9 model, considering the need for rapid convergence and avoiding neuron death, we chose SiLU as the activation function. ReLU may cause some neurons to never be activated during the training process, known as the "death" phenomenon. The Tanh function has smaller gradients when the input is large or small, which can lead to slower training speeds and carries the risk of gradient explosion. The smoothness, gradient retention capability, and computational simplicity of SiLU make it an ideal choice for our model, helping to improve the training efficiency and performance of the model.

## Materials and experiments

In this section, we first introduce the evaluation metrics and experimental details. Then, we report the performance and compare the proposed MAR-YOLOv9 model with existing methods. We conducted a comprehensive assessment of the model's performance using accuracy, recall, F1-score, and other metrics, and analyzed the model's detection results through visualization. Finally, we compared the lightweight nature of the model.

### Experimental conditions and details

In this study, we developed a lightweight, domain-adaptive enhancement object detection method based on the YOLOv9 framework.

To verify the effectiveness and domain adaptability of the proposed method, we validated it on four public plant detection and counting datasets, including the Maize Tassel Detection and Counting UAV (MTDC-UAV) [25], Wheat Ear Detection (WEDU) [26], Rapeseed Flower Rectangle Box Marked Dataset (RFRB) [27], and Rice Panicle Diversity Detection (DRPD) [28]. Below is a detailed introduction to these four datasets:

The Maize Tassel Detection and Counting UAV (MTDC-UAV) dataset is a bounding box annotation dataset for maize tassel detection and counting based on UAV imagery. The dataset has annotated 200 images for the training set and reserved the remaining 106 images for the counting task. To address the issue of spatial information loss due to neural network downsampling, a beneficial technique was employed to increase the number of samples and improve model training efficiency and generalization ability by segmenting the images. Each image was evenly divided into four parts, resulting in a total of 800 images. It should be noted that the corresponding annotation files for the images were also divided into four parts, and bounding boxes exceeding the segmentation size were set back to the segmentation boundary values, similar to annotating partially blurred/visible tassels. By adopting the segmentation method, the feature maps retained more spatial information, which is conducive to precise localization and capturing complex tassel details. Additionally, this method has also been proven to help reduce the demand for computational resources. In summary, the MTDC-UAV dataset includes 500 segmented images for the training set, with the remaining

300 images used for testing detection performance. Moreover, 106 undivided images with point annotations were used to evaluate the model's counting performance.

The Wheat Ear Detection Updated (WEDU) dataset is an extension of the original WEDU dataset initially released by Madec et al. [29], focusing on the precise identification of wheat ears. This updated dataset, while maintaining the advantages of the original dataset, has made significant corrections to the consistency issues between the annotation labels and the images. It contains 165 training images and 71 testing images. It is worth noting that the WEDU dataset cannot completely eliminate the noise that may exist in the annotation boxes, which poses a greater challenge to the model's performance compared to some other well-curated datasets.

The RFRB dataset is a rapeseed flower rectangle box marked dataset collected in the Wuhan area of Hubei, China. The dataset includes 114 rapeseed flower images, with 90 used for training and 24 for testing. These images were taken at a height of 10-15 meters using mobile devices, typically representing the characteristics of UAV imagery, making the dataset highly realistic and practical. It not only simulates the perspective of UAVs in actual farmland monitoring but also captures the complex appearance of rapeseed flowers under different lighting and weather conditions. The number of instances in the dataset ranges widely, from 27 to 629, providing a great challenge for the model when dealing with different object densities. Especially in cases of high object density, the model needs to accurately capture the features of small-scale plants.

The Rice Panicle Diversity Detection (DRPD) dataset is a rice research project that spans multiple geographical regions and varieties, covering experimental fields from various geographical regions such as China, the United States, and Japan, including 229 different rice varieties. Aerial images were taken at three different heights. The dataset contains 200 images for training and 220 images for testing. These images were cropped to a uniform size of $512 \times 512$ pixels. The image quality is relatively low, which presents a challenge for the model to overcome the limitations of low data resolution and blurriness.

In summary, by validating on these four public plant detection and counting datasets, we can comprehensively evaluate the universality and effectiveness of the proposed method. These datasets cover different plant species, growing environments, and image qualities, providing rich data and challenges for the verification of the method. Example images from the four plant datasets are shown in Fig 3.

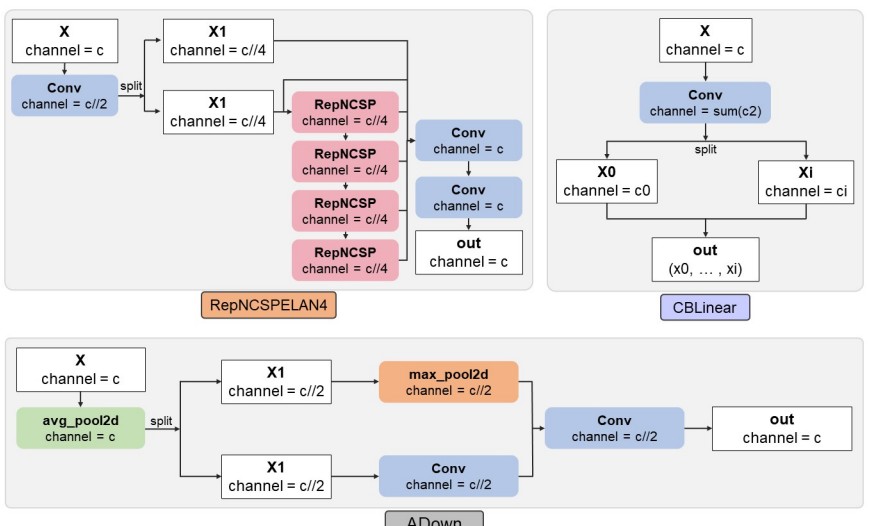

**Fig 3. Example images from four plant datasets.**

To ensure the accuracy and reliability of the experiments, we carefully set up the experimental conditions and refined the experimental details. The experiments were conducted on a machine equipped with an NVIDIA GeForce GTX 3090 GPU, using the PyTorch 2.0.0 deep learning framework [30] for model training and evaluation, with the CUDA version 11.8 parallel computing framework and CUDNN version 8.9.5 deep neural network acceleration library to fully utilize the parallel computing capabilities of the GPU. In the experiments, to make full use of the information in the datasets, standard data preprocessing methods were used, including image scaling, cropping, and normalization, as well as data augmentation techniques such as random flipping and rotation to improve the model's generalization ability. During the model training process, stochastic gradient descent (SGD) was used as the optimizer to avoid the waste of computational resources caused by computing the gradients of the entire dataset. The initial learning rate was set to 0.01, with a batch size of 4 and a momentum factor of 0.937. Training for 300 epochs, based on the consideration of convergence, allowed the model to reach a state of convergence. These experimental conditions and details ensure that the model can fully learn the characteristics of the dataset, for an objective evaluation of the model's performance and accurate comparison. Next, we will verify the effectiveness of the proposed method through comparative experiments and explore its potential and value in practical applications.

## Evaluation metrics

In this subsection, we introduce the metrics used for model evaluation, including Precision (P), Recall (R), and mean Average Precision (mAP), among others. Precision (P) represents the proportion of correctly predicted objects out of all predicted objects, Recall (R) represents the proportion of correctly predicted objects out of all actual objects, and mAP is the average of the precision at various Intersection over Union (IoU) thresholds, providing a more comprehensive assessment of the model's performance across different thresholds. mAP@0.5 and mAP@0.5:0.95 are calculated to evaluate the model's performance at different IoU thresholds. Their definitions are as follows (6)–(8):

$$Precision = \frac{TP}{TP + FP}, \tag{6}$$

$$Recall = \frac{TP}{TP + FN}, \tag{7}$$

$$mAP = \frac{1}{n}\sum_{i=1}^{n}\int_{0}^{1} P_i(R)d(R), \tag{8}$$

where TP (True Positives), FP (False Positives), and FN (False Negatives) represent the number of true positives, false positives, and false negatives, respectively. "TP + FP" is the total number of detected pods, and "TP + FN" is the total number of actual pods in the image. mAP@0.5 indicates the average mAP value when the IoU threshold is 0.5, serving as the main evaluation metric, it can comprehensively assess the model's performance across different IoU thresholds. Additionally, mAP@0.5:0.95 represents the average mAP value at different IoU thresholds (from 0.5 to 0.95, with a step of 0.05), providing a more stringent assessment of performance.

Furthermore, the experiment introduces counting performance metrics such as Mean Absolute Error (MAE), Root Mean Square Error (RMSE), and Coefficient of Determination ($R^2$), allowing for a more comprehensive evaluation of the MAR-YOLOv9 model's

performance in counting tasks. The Mean Absolute Error (MAE) is the average of the absolute differences between predicted and actual values, reflecting the overall error of the model's predictions. The smaller the MAE value, the closer the model's predictions are to the actual values, indicating better counting performance. The Root Mean Square Error (RMSE) is the square root of the average of the squared differences between predicted and actual values, which is very sensitive to large errors, thus reflecting the stability and robustness of the model's predictions. The smaller the RMSE value, the more stable the model's predictions are, indicating more reliable counting performance. The Coefficient of Determination ($R^2$) indicates the degree to which the model fits the data, the closer its value is to 1, the closer the model's predictions are to the actual values, and the stronger the model's explanatory power. A high $R^2$ value not only means good counting performance but also indicates that the model can capture the inherent patterns and trends of the data. Their definitions are as follows (9)–(11):

$$MAE = \frac{1}{n}\sum_{i=1}^{n}|\widehat{y}_i - y_i|, \tag{9}$$

$$RMSE = \sqrt{\frac{1}{n}\sum_{i=1}^{n}(y_i - \widehat{y}_i)^2}, \tag{10}$$

$$R^2 = 1 - \frac{\sum_{i=1}^{n}(y_i - \widehat{y}_i)^2}{\sum_{i=1}^{n}(y_i - \bar{y}_i)^2}, \tag{11}$$

## Comparison of model performance with different object detection methods

In this study, to comprehensively evaluate the performance of the proposed object detection model, we selected multiple evaluation metrics and compared them with benchmark models on four public datasets. These benchmark models include CenterNet [9], Faster R-CNN [8], Efficientdet [31], DETR [32], FCOS [33], SSD [4], YOLOv8 [34], and YOLOv9 [5]. MAR-YO-LOv9 was trained and tested under the same experimental conditions as these benchmark models. Through experiments, the performance of these models was compared and analyzed in terms of accuracy (Precision, P), recall (Recall, R), and mean Average Precision (mAP). It was evident that MAR-YOLOv9 demonstrated superior performance across different datasets. On the average mAP@0.5 of the four datasets, compared with seven mainstream algorithms (CenterNet, Faster R-CNN, Efficientdet, DETR, FCOS, SSD, YOLOv8), MAR-YOLOv9 achieved an average accuracy improvement of 39.18%. Specifically, on the MTDC-UAV, WEDU, RFRB, and DRPD datasets, the mAP@0.5 values of MAR-YOLOv9 increased by 1.2%, 0.8%, 1.8%, and 1.3% compared to YOLOv9, respectively (with an average of 1.28%). It also showed comparable performance in the more stringent mAP@0.5:0.95 metric, with steady progress. CenterNet, with its anchor-free design, simplifies the detection process but may have limitations in detecting overlapping objects and keypoint matching. Efficientdet may sacrifice some accuracy in pursuit of efficiency, especially in small object detection. It is worth noting that DETR, as a Transformer-based model, differs from other object detection models, employing an end-to-end training approach that reduces cumbersome steps and human intervention. However, it is easily limited by labeled data and computational resources, and its high computational cost makes it unsuitable for real-time application scenarios, where it did not perform well. Experiments show that MAR-YOLOv9 maintains high-precision detection while also having stronger generalization capabilities, maintaining stable performance across

different datasets. Tables 1–4 below show the quantitative results of each model on the four datasets.

**Model performance evaluation compared to different object detection methods.** After evaluating the performance of different models, assessing the performance in counting tasks is crucial for the model's accuracy and reliability in practical applications. By comparing the MAR-YOLOv9 model with benchmark models in counting tasks, we can evaluate the consistency between predicted and actual values. The experimental results are shown in Tables 5–8.

**Table 1. Quantitative results on MTDC-UAV dataset.**

| Model | P | R | mAP@0.5 | mAP@0.5:0.95 |
|---|---|---|---|---|
| CenterNet | 0.802 | **0.858** | 81.4% | 20.6% |
| Faster R-CNN | 0.327 | 0.273 | 16.1% | 4.3% |
| Efficientdet | 0.376 | 0.430 | 30.4% | 5.6% |
| DETR | 0.012 | 0.010 | 7.0% | 0.1% |
| FCOS | **0.883** | 0.580 | 66.6% | 26.7% |
| SSD | 0.372 | 0.144 | 16.5% | 3.4% |
| YOLOv8 | 0.815 | 0.717 | 76.3% | 32.3% |
| YOLOv9 | 0.847 | 0.768 | 82.2% | 38.5% |
| MAR-YOLOv9 | 0.852 | 0.784 | **83.4%** | **39.7%** |

The best performance is highlighted in bold.

**Table 2. Quantitative results on WEDU dataset.**

| Model | P | R | mAP@0.5 | mAP@0.5:0.95 |
|---|---|---|---|---|
| CenterNet | 0.768 | 0.534 | 46.5% | 19.7% |
| Faster R-CNN | 0.454 | 0.583 | 52.1% | 19.9% |
| Efficientdet | 0.408 | 0.427 | 41.7% | 17.8% |
| DETR | 0.198 | 0.119 | 10.3% | 2.1% |
| FCOS | 0.574 | 0.656 | 62.6% | 25.5% |
| SSD | 0.721 | 0.542 | 61.1% | 23.2% |
| YOLOv8 | 0.898 | 0.844 | 91.0% | 54.5% |
| YOLOv9 | 0.933 | 0.911 | 95.7% | 63.9% |
| MAR-YOLOv9 | **0.939** | **0.916** | **96.5%** | **65.1%** |

**Table 3. Quantitative results on RFRB dataset.**

| Model | P | R | mAP@0.5 | mAP@0.5:0.95 |
|---|---|---|---|---|
| CenterNet | 0.791 | 0.590 | 50.2% | 21.1% |
| Faster R-CNN | 0.463 | 0.463 | 46.3% | 18.5% |
| Efficientdet | 0.195 | 0.254 | 10.4% | 0.9% |
| DETR | 0.081 | 0.195 | 2.4% | 0.5% |
| FCOS | 0.913 | 0.404 | 77.3% | 10.9% |
| SSD | 0.183 | 0.477 | 8.0% | 0.3% |
| YOLOv8 | 0.886 | **0.835** | **90.6%** | 48.2% |
| YOLOv9 | 0.923 | 0.732 | 83.6% | 48.1% |
| MAR-YOLOv9 | **0.934** | 0.750 | 85.4% | **50.9%** |

**Table 4. Quantitative results on DRPD dataset.**

| Model | P | R | mAP@0.5 | mAP@0.5:0.95 |
|---|---|---|---|---|
| CenterNet | 0.830 | **0.944** | 94.2% | 39.8% |
| Faster R-CNN | 0.549 | 0.677 | 54.9% | 16.5% |
| Efficientdet | 0.400 | 0.264 | 24.8% | 6.6% |
| DETR | 0.334 | 0.806 | 57.2% | 22.2% |
| FCOS | 0.887 | 0.887 | 89.3% | 50.4% |
| SSD | 0.689 | 0.629 | 69.7% | 29.6% |
| YOLOv8 | 0.877 | 0.863 | 92.2% | 57.9% |
| YOLOv9 | 0.899 | 0.898 | 94.0% | 63.6% |
| MAR-YOLOv9 | **0.918** | 0.900 | **95.3%** | **67.0%** |

**Table 5. Counting results on MTDC-UAV dataset.**

| Model | MAE | RMSE | $R^2$ |
|---|---|---|---|
| CenterNet | 21.20 | 29.53 | 0.9380 |
| Faster R-CNN | 103.78 | 134.03 | 0.3012 |
| Efficientdet | 28.51 | 39.56 | 0.9011 |
| DETR | 201.35 | 277.51 | 0.5790 |
| FCOS | 31.32 | 40.11 | 0.9096 |
| SSD | 35.77 | 47.27 | 0.8444 |
| YOLOv8 | 26.99 | 39.63 | 0.9004 |
| YOLOv9 | 21.16 | 30.82 | 0.9330 |
| MAR-YOLOv9 | **19.75** | **27.42** | **0.9478** |

**Table 6. Counting results on WEDU dataset.**

| Model | MAE | RMSE | $R^2$ |
|---|---|---|---|
| CenterNet | 26.86 | 38.73 | 0.8971 |
| Faster R-CNN | 28.90 | 36.11 | 0.8732 |
| Efficientdet | 72.34 | 106.40 | 0.3264 |
| DETR | 116.30 | 186.61 | 0.2362 |
| FCOS | 53.13 | 69.13 | 0.7149 |
| SSD | 30.24 | 40.19 | 0.7636 |
| YOLOv8 | 7.83 | 10.96 | 0.9145 |
| YOLOv9 | 5.89 | 8.53 | 0.9374 |
| MAR-YOLOv9 | **5.87** | **7.66** | **0.9513** |

Analyzing the experimental results of counting performance, MAR-YOLOv9 also demonstrated excellent performance. Across the four datasets, the MAE and RMSE values of MAR-YOLOv9 were lower than those of the other models. Compared to YOLOv9, the MAE values of MAR-YOLOv9 decreased by 1.41, 0.02, 2.16, and 0.15 on the MTDC-UAV, WEDU, RFRB, and DRPD datasets, respectively, and the RMSE values decreased by 3.40, 0.87, 8.85, and 0.19, respectively. Notably, the counting performance improvement of MAR-YOLOv9 on the RFRB dataset was more pronounced. We attribute this mainly to MAR-YOLOv9's strong adaptability to changes in density. The RFRB dataset contains rapeseed flowers with a wide range of densities, from low to high, and this variation in density poses a challenge to the model's

**Table 7. Counting results on RFRB dataset.**

| Model | MAE | RMSE | $R^2$ |
|---|---|---|---|
| CenterNet | 25.54 | 34.26 | 0.9541 |
| Faster R-CNN | 137.33 | 173.84 | 0.5649 |
| Efficientdet | 39.96 | 50.03 | 0.9015 |
| DETR | 275.50 | 363.68 | 0.5141 |
| FCOS | 35.67 | 52.28 | 0.9036 |
| SSD | 76.12 | 100.22 | 0.8282 |
| YOLOv8 | 26.67 | 37.87 | 0.9418 |
| YOLOv9 | 27.04 | 39.54 | 0.9471 |
| MAR-YOLOv9 | **24.88** | **30.69** | **0.9593** |

**Table 8. Counting results on DRPD dataset.**

| Model | MAE | RMSE | $R^2$ |
|---|---|---|---|
| CenterNet | 2.48 | 3.12 | 0.9067 |
| Faster R-CNN | 2.72 | 3.49 | 0.8226 |
| Efficientdet | 3.45 | 4.23 | 0.7527 |
| DETR | 4.00 | 4.99 | 0.7303 |
| FCOS | 2.04 | 2.69 | 0.9185 |
| SSD | 4.15 | 5.58 | 0.5517 |
| YOLOv8 | 2.05 | 2.71 | 0.9012 |
| YOLOv9 | 1.63 | 2.16 | 0.9334 |
| MAR-YOLOv9 | **1.48** | **1.97** | **0.9422** |

recognition capabilities. Moreover, the RFRB dataset has more targets than the other three datasets, making it more sensitive to changes in counting performance. MAR-YOLOv9, with its improved structure, successfully overcame this challenge, achieving accurate detection of plants at different densities.

Considering the characteristics of plant detection tasks, our selection of the four datasets is representative as they include plant images of varying scales, densities, resolutions, clarity, and lighting conditions, which allows for a more objective evaluation of the models. Upon comprehensive analysis of the performance of various models across different datasets, it can be observed that poor performance is often closely related to certain aspects of model design. CenterNet underperforms in detecting small objects against complex backgrounds, which may be related to its anchor box design or the structure of its feature pyramid network, as well as simplifying the object detection task by representing targets as center points, leading to the model not accurately capturing the details of small-scale targets. For the two-stage detection method Faster R-CNN, which typically should offer high detection accuracy, did not show a significant advantage in this experiment, with a notably lower recall rate, which may be related to the performance of the Region Proposal Network (RPN). DETR exhibited poor performance across all datasets, which we believe may be due to the challenges its Transformer-based detection framework faces when dealing with fine-grained plant detection tasks. YOLOv8, despite having a balance in Precision (P) and Recall (R), had a lower overall mean Average Precision (mAP), which may be related to the depth, width, or design of its detection head. While YOLOv9 performed well in most metrics, MAR-YOLOv9 still achieved further performance improvements on this basis.

In summary, by comprehensively considering both object detection and counting performance, we have verified the superior comprehensive performance of the MAR-YOLOv9 model. It has achieved improvements to varying degrees in all metrics and demonstrated better performance across different datasets, accurately marking the positive class. It is noteworthy that MAR-YOLOv9 still maintains high performance under the more stringent evaluation condition of mAP@0.5:0.95, reflecting its stronger domain adaptability.

## Visualization and typical error analysis

To gain a deeper understanding of the performance differences between YOLOv9 and MAR-YOLOv9 in object detection tasks, and how these differences lead to counting errors, we conducted a detailed comparative analysis of their visualization results. Through this process, we aim to identify specific types of targets where the models are prone to errors, providing targeted guidance for subsequent model optimization.

Firstly, when setting the confidence threshold, we fully considered the model's performance in achieving the best counting metrics across the entire dataset. This approach ensures that we can objectively evaluate the model's overall performance while filtering out the model's more confident detection results, thus avoiding the impact of low-confidence predictions on the assessment results. Subsequently, we conducted detections on the test images of the dataset and output the corresponding results. To present these results more intuitively, we selected representative error types for visualization, as shown in Figs 4–6. These images not only showcase the model's performance in different scenarios but also highlight the common errors made by the model on specific target types.

Through careful observation of these results, we identified the following typical errors and their causes:

Object Omissions and False Detections: In the MTDC-UAV dataset, YOLOv9 exhibited a noticeable tendency to miss small, occluded, or targets with similar background colors. This is mainly because the model may not fully capture the detailed information of these small targets during feature extraction, leading to their neglect in subsequent detection stages. In contrast, MAR-YOLOv9 performed better in handling such targets. Similar issues of missed detections on occluded wheat ears and false detections of bright leaves as wheat ears in the WEDU dataset also exist. MAR-YOLOv9 improved the model's recognition accuracy by enhancing the loss function and introducing more refined feature fusion strategies, reducing the occurrence of false and missed detections.

Bounding Box Issues: In some detection results, we noticed inaccurate bounding box regression by YOLOv9. For example, in the WEDU dataset, only half of the wheat ears detected by YOLOv9 were enclosed within the bounding box, affecting the model's localization accuracy. This is usually due to inaccurate bounding box regression by the model, resulting in biases when generating prediction boxes. In comparison, MAR-YOLOv9 performed more admirably in these issues.

Detection of Small and Dense Objects: In the RFRB dataset, some targets were closely connected, posing a challenge for detection. However, YOLOv9 tended to enclose such targets together, leading to inaccurate counting. MAR-YOLOv9, by introducing multi-scale feature fusion and more refined anchor box design, enhanced the model's ability to handle dense targets, better separating them.

Detection Performance in Low-Quality Images: The low resolution, poor quality, and uneven lighting conditions of images pose a severe challenge to object detection tasks. However, MAR-YOLOv9 still maintained good performance in such complex environments,

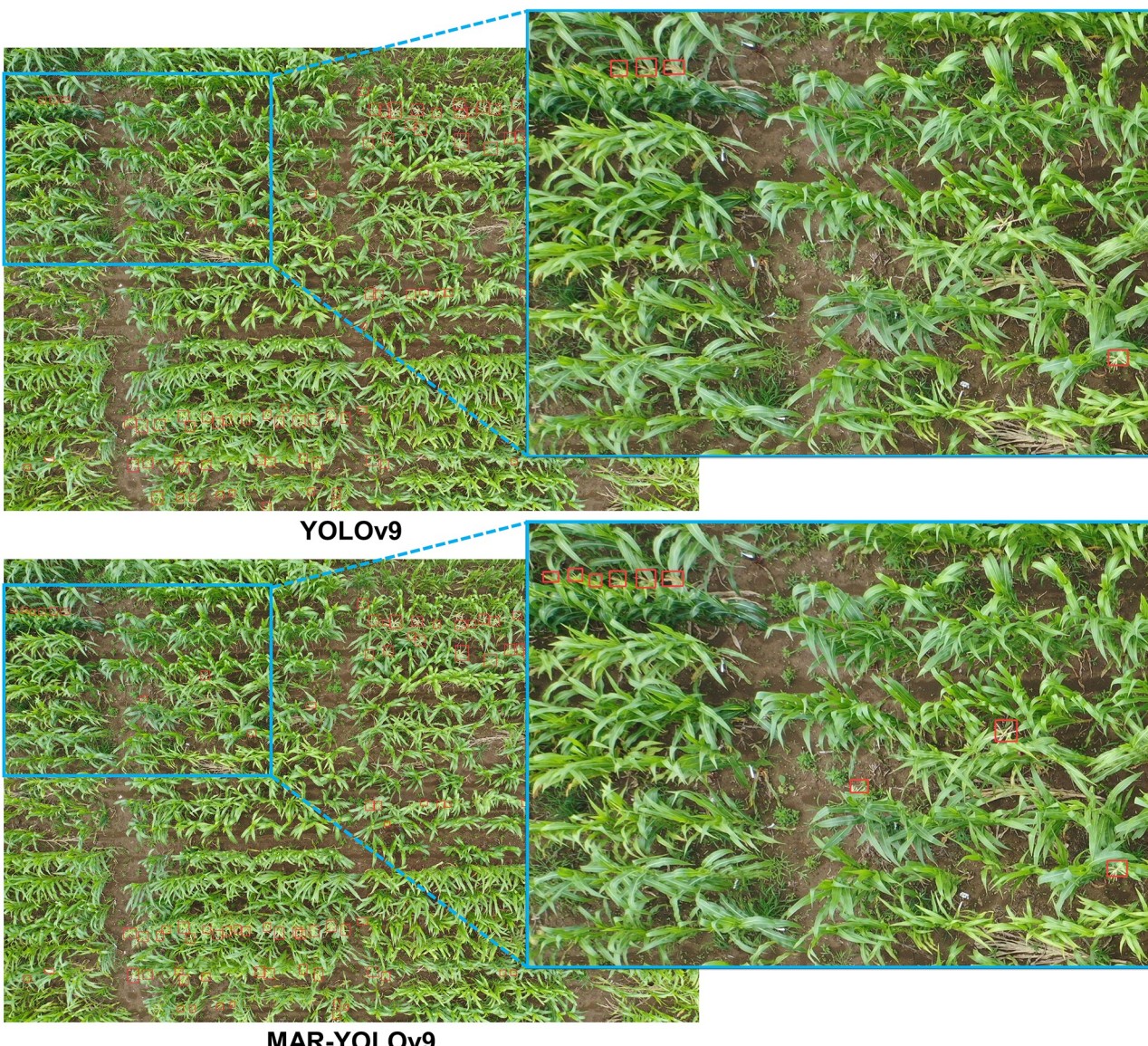

**Fig 4. Detection visualization results on MTDC-UAV dataset.**

successfully identifying rice panicles. This result indicates that MAR-YOLOv9 has strong robustness when dealing with low-quality images.

In summary, through a detailed analysis of the visualization results of YOLOv9 and MAR-YOLOv9, MAR-YOLOv9 demonstrated strong robustness and excellent performance. At the same time, we can better understand the causes of counting errors, which will provide us with powerful guidance for subsequent model optimization, further improving the model's performance in fruit object detection tasks in the agricultural field.

## Model lightweighting

Model lightweighting is an important research direction in the field of deep learning, especially in application scenarios with limited resources, such as the deployment of edge devices in the

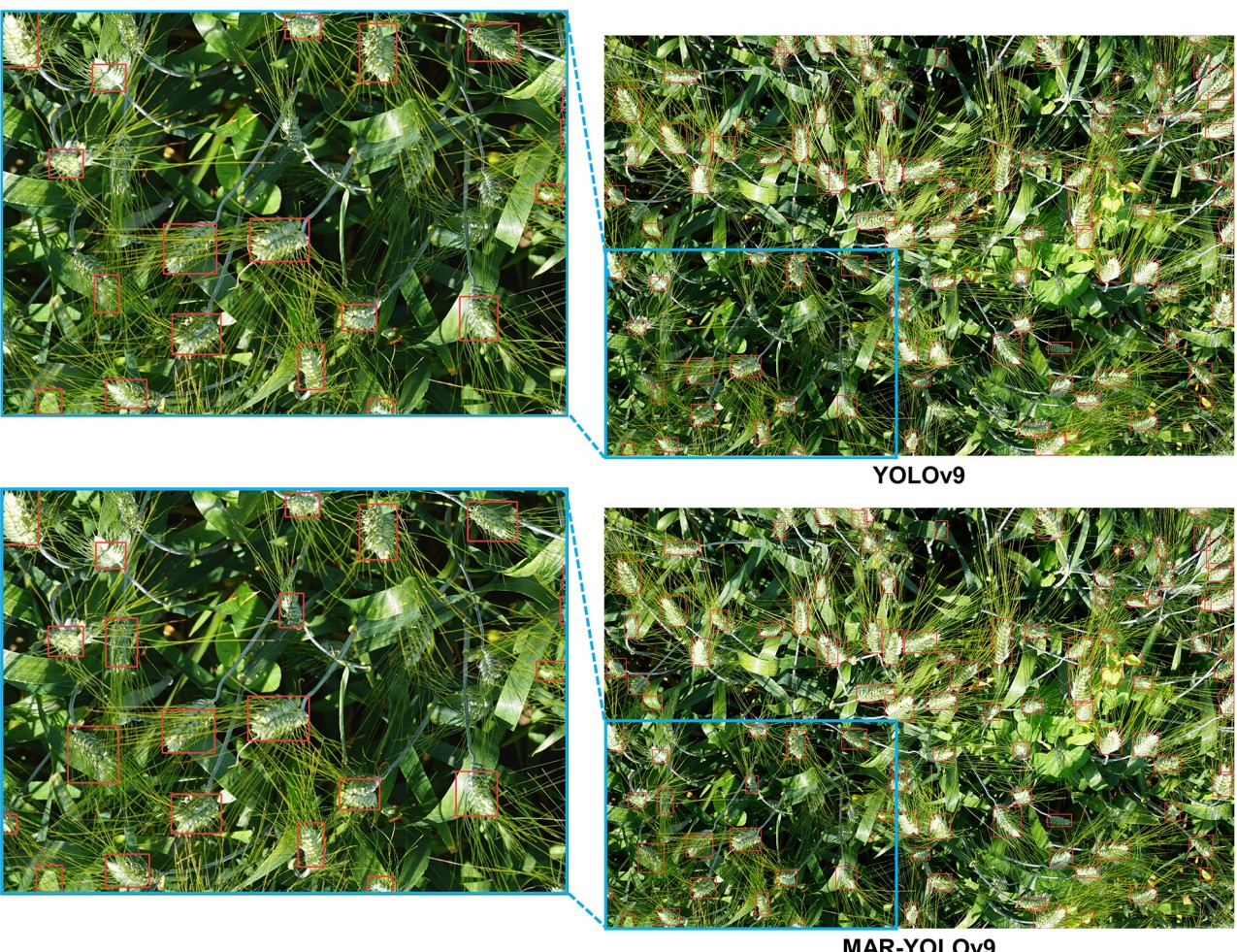

**Fig 5. Detection visualization results on WEDU dataset.**

agricultural field. Although YOLOv8 leads in lightweight metrics with its extremely small model size (6.0MB) and shorter inference time (0.020 seconds), we cannot ignore the performance of the model in the pursuit of lightweighting. According to the count results from previous experiments, the performance of YOLOv8 is not as good as YOLOv9 and MAR-YOLOv9. Notably, MAR-YOLOv9 maintains high precision and stability while implementing a series of optimization measures to achieve model lightweighting. The model size is reduced by 9.3% from YOLOv9's 98.0MB to 88.9MB, and the number of model layers is also decreased from 724 to 550, simplifying the model complexity and further enhancing inference speed. In Fig 7, a chart is used to show the comparison of average inference time and model size on four datasets for different models. The reduction in model parameters and layers not only means that the model occupies less storage space and simplifies the model complexity, but also effectively reduces the computing resources required by the model. In the agricultural field, considering the concern of most farmers for cost-effectiveness, seeking a more economical solution is particularly important. Such a lightweight design of MAR-YOLOv9 is suitable for deployment on edge devices with limited resources and can be promoted for use in the agricultural field.

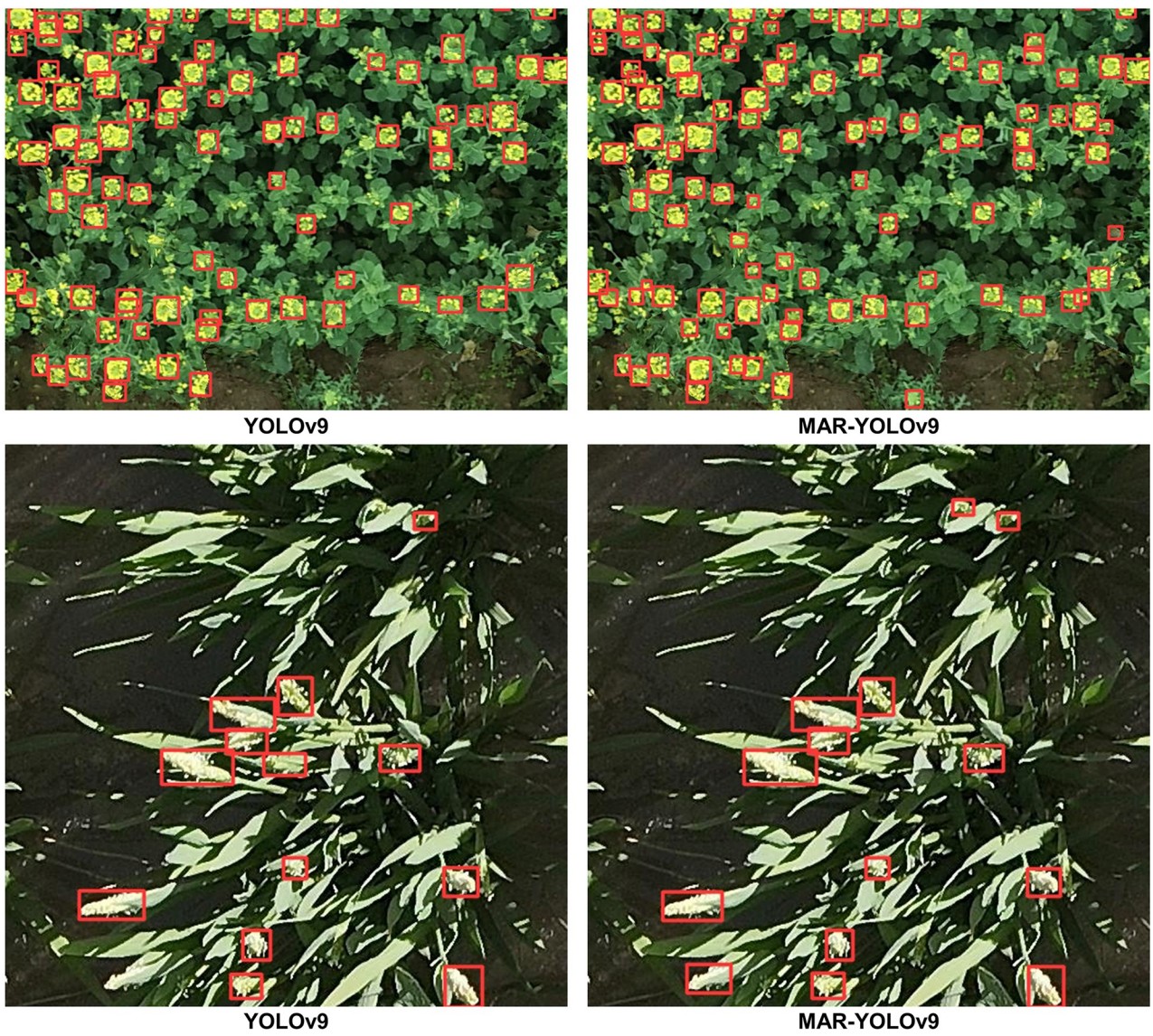

**Fig 6. Detection visualization results on RFRB and DRPD datasets.**

## Discussion

In this paper, we focused on the domain adaptability of the lightweight MAR-YOLOv9 model across four different datasets and validated its advantages in plant object detection tasks through comparative analysis. The research results indicate that MAR-YOLOv9 not only achieves model lightweighting but also demonstrates strong domain adaptability, providing robust support for applications in the agricultural field.

Firstly, by conducting experiments on four datasets with varying scales, densities, resolutions, clarity, and lighting conditions, we comprehensively evaluated the performance of MAR-YOLOv9. This diverse selection of datasets allows us to objectively assess the model's domain adaptability, ensuring its ability to handle various complex scenarios in practical applications. Experimental results show that MAR-YOLOv9 has achieved significant improvements

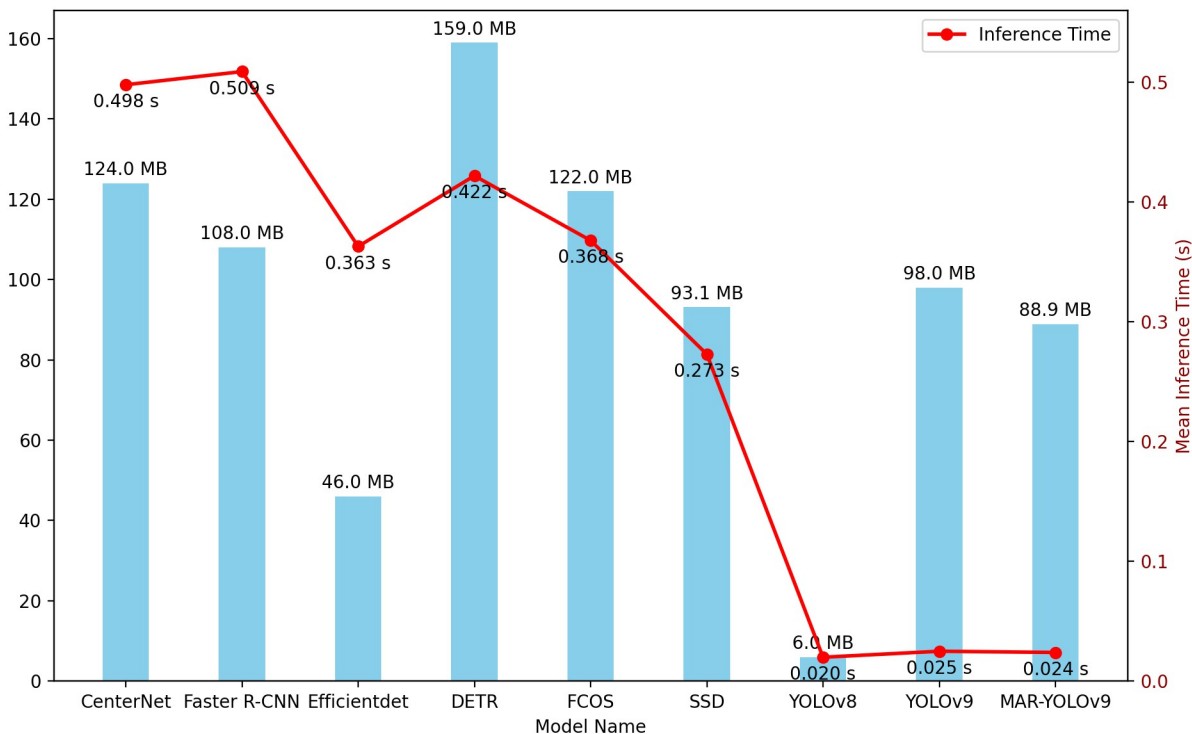

**Fig 7. Performance and model size comparison of object detection models.**

in all metrics, demonstrating superior performance. Secondly, the lightweight design of MAR-YOLOv9 provides great convenience for its application in the agricultural field. Compared to other models, MAR-YOLOv9 significantly reduces model parameters and layers while maintaining high accuracy and stability, reducing storage and computational costs. This enables MAR-YOLOv9 to be easily deployed on edge devices with limited resources, providing real-time and accurate detection services for agricultural production. Additionally, the superior performance of MAR-YOLOv9 is also attributed to its unique design features. The model adopts a lightweight Convolutional Neural Network (CNN) structure, feature fusion technology, and a reversible auxiliary branch, effectively improving detection accuracy and stability. At the same time, MAR-YOLOv9 also optimizes for challenges such as small and dense targets, further enhancing its high adaptability in the agricultural field.

However, when discussing the application achievements of MAR-YOLOv9 in agricultural object detection, although significant progress has been made, we also identified some existing shortcomings and areas for improvement. Especially in the visualization analysis, we observed occasional false negatives and false positives with MAR-YOLOv9, mainly in situations where fruits are occluded, the background is complex, or there are significant lighting changes. These types of errors affect the model's performance and prevent it from reaching a higher level. These situations pose challenges to the model's performance and also point out directions for further optimization.

## Conclusion

The aim of this study was to address the issue of domain adaptability in agricultural object detection by developing the MAR-YOLOv9 model based on the YOLOv9 framework,

achieving significant performance improvements. The model innovatively designed a 16x downsampled Backbone network and improved the main detection neck, enhancing the model's adaptability in agricultural scenarios. In the experiments, four different datasets were used for comprehensive evaluation, and the results showed that MAR-YOLOv9 achieved higher levels in metrics such as precision, recall, and mAP compared to other benchmark models. Especially in scenarios with occluded fruits, complex backgrounds, and significant lighting changes, MAR-YOLOv9 also demonstrated strong robustness. At the same time, MAR-YO-LOv9 has a smaller number of parameters and model layers, reducing storage and computational costs while maintaining high accuracy. This feature makes MAR-YOLOv9 easier to deploy on edge devices with limited resources, providing possibilities for real-time object detection in the agricultural field. Future research can build on the basis of this study to further explore how to improve and optimize the object detection model by optimizing network structures, introducing more contextual information, or utilizing transfer learning methods. At the same time, we will also focus on more practical needs in agricultural applications, such as improving the model's recognition capabilities for specific crops and achieving simultaneous detection of multiple targets, to meet the diverse object detection needs in agricultural production.

## Author Contributions

**Conceptualization:** Dunlu Lu, Yangxu Wang.

**Data curation:** Dunlu Lu, Yangxu Wang.

**Formal analysis:** Yangxu Wang.

**Funding acquisition:** Dunlu Lu.

**Investigation:** Dunlu Lu.

**Methodology:** Dunlu Lu, Yangxu Wang.

**Project administration:** Dunlu Lu.

**Resources:** Dunlu Lu.

**Software:** Yangxu Wang.

**Supervision:** Dunlu Lu.

**Validation:** Dunlu Lu, Yangxu Wang.

**Visualization:** Yangxu Wang.

**Writing – original draft:** Yangxu Wang.

**Writing – review & editing:** Dunlu Lu.

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
