## [Decision Letter · Decision Letter 0]

1 May 2024

PONE-D-24-14684MAR-YOLOv9: An Agricultural Domain Adaptive Object Detection Method for Unmanned Aerial Vehicles Based on YOLOv9PLOS ONE

Dear Dr. Wang,

Thank you for submitting your manuscript to PLOS ONE. After careful consideration, we feel that it has merit but does not fully meet PLOS ONE’s publication criteria as it currently stands. Therefore, we invite you to submit a revised version of the manuscript that addresses the points raised during the review process.

We look forward to receiving your revised manuscript.

Kind regards,

Xiaoyong Sun

Academic Editor

PLOS ONE

“This work was funded in part by 2022 key scientific research project of ordinary universities in Guangdong Province under Grant 2022ZDZX4075, in part by 2022 Guangdong province ordinary universities characteristic innovation project under Grant 2022KTSCX251, in part by the 2021 Guangdong Provincial Continuing Education Quality Improvement Project under Grant JXJYGC2021CY0173, in part by the Collaborative Intelligent Robot Production & Education Inte-grates Innovative Application Platform Based on the Industrial Internet under Grant 2020CJPT004, in part by 2020 Guangdong Rural Science and Technology Mission Project under Grant KTP20200153, in part by the Engineering Research Center for Intelligent equipment manufac-turing under Grant 2021GCZX018.”

5. We note that Figures 1, 4, 5 and 6 in your submission contain copyrighted images. All PLOS content is published under the Creative Commons Attribution License (CC BY 4.0), which means that the manuscript, images, and Supporting Information files will be freely available online, and any third party is permitted to access, download, copy, distribute, and use these materials in any way, even commercially, with proper attribution. For more information, see our copyright guidelines: http://journals.plos.org/plosone/s/licenses-and-copyright.

1. You may seek permission from the original copyright holder of Figures 1, 4, 5 and 6 to publish the content specifically under the CC BY 4.0 license.

Reviewers' comments:

Reviewer's Responses to Questions

**Comments to the Author**

1. Is the manuscript technically sound, and do the data support the conclusions?

Reviewer #1: Yes

Reviewer #2: Yes

2. Has the statistical analysis been performed appropriately and rigorously? 

Reviewer #1: Yes

Reviewer #2: Yes

3. Have the authors made all data underlying the findings in their manuscript fully available?

Reviewer #1: Yes

Reviewer #2: Yes

4. Is the manuscript presented in an intelligible fashion and written in standard English?

Reviewer #1: Yes

Reviewer #2: Yes

5. Review Comments to the Author

Reviewer #1: (1) Title：

- It should be noted that the paper in question is not part of a study that addresses domain adaptation, nor does it make use of any techniques related to domain adaptation. Consequently, the full paper must be revised.

(2) Abstract：

- In order to emphasise the innovative aspect of the proposed methodology, it is necessary to provide a more detailed description of the challenges that the problem presents.

- The authors concluded that the method improves performance while reducing the computational complexity of the model and speeding up the detection process. However, there is a lack of experimental data to substantiate this assertion.

(3) Introduction:

- The research presented in this paper is based on older studies, and it would be beneficial to include more recent research in the discussion.

- The description of the challenge of the problem and the motivation for the research that is presented in this paper is not sufficiently clear. It is therefore necessary to provide greater clarity.

- The paper is deficient in the inclusion of a section on related work.

- The contribution of this paper could be enhanced by emphasising the innovative aspect of the proposed solution. Simply replacing the 32x downsampling backbone with a 16x downsampling backbone does not constitute a significant advancement.

(4) Materials and Methods:

- It should be noted that the subsection "Datasets" is not part of the proposed methodology of this paper and should therefore appear in the section "Experiment".

- It is recommended that a description of the agricultural image pre-processing be included, along with an explanation of the methods employed by the authors to make YOLOv9 applicable to the field.

(5) Experiment：

- It is recommended that assessment indicators not be described in a fragmented manner. It is in the reader's interest to have them presented in a separate subsection.

- The current methodologies employed in experimental comparisons are inadequate and require augmentation.

Reviewer #2: This paper proposes a MAR-YOLOv9 model based on the YOLOv9 framework to automatically detect crops. This model takes into account both domain adaptability and lightweight requirements, and makes some improvements and innovations in the design of neural network modules. In the comparative experiments of four plant data sets, the superiority of this method is demonstrated. It is recommended to accept after the following modifications:

1. The abstract only gives the superiority between this model and YOLOv9, and lacks a description of the comparison with other comparative algorithms. It is recommended to add.

2. The Introduction has explained the background of this paper and the purpose of the algorithm. The content of lines 168-174 in the Model construction part seems a bit lengthy. It is recommended to simplify it.

3. For some reason, the pictures Fig1-Fig6 in the article seem to be compressed, not clear enough, and color deviation occurs.

4. In the Backbone and Neck parts, there are no references and simple descriptions for modules such as Silence, SPPELAN, and CBFuse. It is recommended that the author add.

5. In the Activation Functions part, the author lists three different activation functions and gives a brief description, but finally chooses one of them to build his own network. Therefore, the content of this section is a bit redundant, please consider deleting it as appropriate.

6. It is recommended that the English expression of the paper be further optimized and improved to facilitate readers' understanding.

6. PLOS authors have the option to publish the peer review history of their article (what does this mean?). If published, this will include your full peer review and any attached files.

Reviewer #1: No

Reviewer #2: No

---

## [Author Response · Author response to Decision Letter 0]

3 Jun 2024

Dear Reviewers: 

Thank you for your letter and for the reviewers' comments concerning our manuscript entitled “MAR-YOLOv9: An Agricultural Domain Adaptive Object Detection Method for Unmanned Aerial Vehicles Based on YOLOv9”. The title of the article has now been revised to "MAR-YOLOv9: A Multi-Dataset Object Detection Method for Agricultural Fields Based on YOLOv9". Those comments are all valuable and very helpful for revising and improving our paper, as well as the important guiding significance to our researches. We have carefully considered the comments and have made the necessary corrections, hoping to meet with your approval. Due to the language polishing, extensive revisions have been made to the article.

To Reviewer 1:

We are truly grateful to you for your profound comments and valuable suggestions.

1. Title: It should be noted that the paper in question is not part of a study that addresses domain adaptation, nor does it make use of any techniques related to domain adaptation. Consequently, the full paper must be revised.

Response: Thank you for your feedback. Our study indeed does not involve traditional domain adaptation techniques but rather employs a method of multi-dataset evaluation. In the agricultural field, the challenges we face include a wide variety of crops and changing environmental conditions. Therefore, we have proposed a model that is trained and evaluated separately on multiple different datasets within the agricultural domain. This approach allows our model to better understand and adapt to the diversity within the agricultural field rather than transferring from one domain to another.

Our method is more closely related to multi-dataset evaluation, aiming to assess and enhance the model's generalization capabilities and performance across different subsets of the agricultural domain, such as different crop types, various growth stages, or different environmental conditions. We changed the title of the paper to "MAR-YOLOv9: A Multi-Dataset Object Detection Method for Agricultural Fields Based on YOLOv9" and made appropriate changes to the content of the paper to more accurately reflect our research methodology and focus. Thank you once again for your valuable comments.

2. Abstract: In order to emphasise the innovative aspect of the proposed methodology, it is necessary to provide a more detailed description of the challenges that the problem presents.

Response: Thank you very much for your attention to our research and your valuable suggestions. In response to the issues you raised, we have made corresponding improvements in the abstract. Specifically, in the design of the Neck, the hybrid connection strategy allows the model to flexibly utilize features from different levels. However, due to the limited space in the abstract, we have endeavored to streamline our expression to highlight the challenges of the problem and the innovative aspects of our approach. Once again, we are grateful for your valuable feedback.

3. Abstract: The authors concluded that the method improves performance while reducing the computational complexity of the model and speeding up the detection process. However, there is a lack of experimental data to substantiate this assertion.

Response: Your observations are indeed pertinent. Addressing the issue you pointed out regarding the lack of experimental data to substantiate the performance enhancement and reduction in computational complexity of our model, we have made the appropriate additions and corrections. In the latest revision, we have included Figure 7, which provides a comparison of specific inference times and model sizes, along with an analysis of the results. By comparing the inference times of different models, we are able to more concretely demonstrate the advantages of our proposed model in terms of reducing computational complexity and accelerating the detection process. We believe that this data will strongly support the relevant conclusions presented in our paper.

4. Introduction: The research presented in this paper is based on older studies, and it would be beneficial to include more recent research in the discussion.

Response: Thank you for your valuable comments. We understand your concern about including more recent studies in our work. We confirm that in the section of related work, our previous manuscript has already cited and discussed several studies from after 2020, which are all recent advancements within the past four years, particularly in the areas of object detection and its applications in agriculture. Furthermore, we have incorporated discussions on the latest research findings from the past two years to ensure that our work remains at the forefront of scientific research.

5. Introduction: The description of the challenge of the problem and the motivation for the research that is presented in this paper is not sufficiently clear. It is therefore necessary to provide greater clarity.

Response: Thank you very much for your professional review and constructive suggestions! We have supplemented the content in the introduction section of our paper to provide a richer description of the challenges and research motivations.

6. Introduction: The paper is deficient in the inclusion of a section on related work.

Response: Your insights are truly perceptive. In the original manuscript, the section on related work was integrated within the introduction. Now, we have separated it into a distinct subsection following the introduction chapter, and have also addressed the issue of including the most recent literature as you previously mentioned. We hope that the revised version meets your requirements.

7. Introduction: The contribution of this paper could be enhanced by emphasising the innovative aspect of the proposed solution. Simply replacing the 32x downsampling backbone with a 16x downsampling backbone does not constitute a significant advancement.

Response: Thank you very much for your feedback. In the latest version of the paper, we have placed greater emphasis on the innovative aspects of the proposed solution. The innovation of our model is not merely the substitution of a 32x downsampling backbone with a 16x downsampling backbone. More importantly, in the decoding layer, we have innovatively integrated multi-scale feature fusion, a reversible auxiliary branch, and a feature map fusion module. This hybrid connection strategy allows the model to flexibly utilize features from different levels and produce richer feature representations. This design not only improves the model's detection performance for objects but also enhances the model's robustness and computational efficiency. In comparative experiments, we have fully demonstrated the advantages of the innovative connection method in improving model performance.

We firmly believe that these revisions and additions better reflect the innovation and contributions of this paper. We sincerely hope that these modifications meet your expectations and look forward to your further guidance and suggestions.

8. Materials and Methods: It should be noted that the subsection "Datasets" is not part of the proposed methodology of this paper and should therefore appear in the section "Experiment".

Response: Thank you for bringing up this important suggestion. Indeed, based on the structure of the paper, the section on "Datasets" would be better placed within the "Experiments" section. We have made the adjustment in the new version accordingly.

9. Materials and Methods: It is recommended that a description of the agricultural image pre-processing be included, along with an explanation of the methods employed by the authors to make YOLOv9 applicable to the field.

Response: Thank you for your meticulous review and suggestions regarding the paper. In response to the issues you raised about agricultural image preprocessing, I would like to provide further clarification on our research methodology. In our study, we utilized a pre-processed agricultural image dataset directly. To ensure the fairness and comparability of our research, we did not perform any additional preprocessing or data augmentation on the original dataset. Our work focused solely on the application of the model, where we made corresponding adjustments and optimizations to the MAR-YOLOv9 model to cater to the characteristics of targets in agricultural images. We understand that this approach may limit the model's performance enhancement in some aspects, but it avoids additional data enhancement or processing to maintain the fairness of the experiments and the comparability of the results. We acknowledge your concerns about the data preprocessing component, and in future work, our research can continue to explore more agricultural image preprocessing methods to further enhance the model's performance.

10. Experiment: It is recommended that assessment indicators not be described in a fragmented manner. It is in the reader's interest to have them presented in a separate subsection.

Response: Thank you for your valuable suggestions. We have made adjustments to the experimental section of the article, concentrating the description of the evaluation metrics into a separate subsection. In this new subsection, we have provided a detailed explanation of the definitions, calculation methods, and the importance of these evaluation metrics in assessing the performance of object detection models. By organizing the content in this manner, we hope to help readers better understand our experimental processes and results, and to enhance the flow and readability of the article. We believe that these modifications will improve the quality of the paper and the reader's experience.

11. Experiment: The current methodologies employed in experimental comparisons are inadequate and require augmentation.

Response: Thank you for pointing out the issue. In response to your suggestion, we have added new comparative models in our experiments, including Efficientdet, DETR, FCOS, and SSD, bringing the total number of comparison methods to eight. Additionally, we have included their experimental results in the comparative experimental tables, Tables 1-8. By introducing these new models, we hope to provide a more comprehensive and objective assessment of the performance merits of our proposed model. At the same time, we have conducted an in-depth analysis of the experimental results to better understand the performance of different models in various scenarios.

To Reviewer 2:

Thank you for your valuable comments, which have helped us to substantially improve the manuscript.

1. The abstract only gives the superiority between this model and YOLOv9, and lacks a description of the comparison with other comparative algorithms. It is recommended to add.

Response: Thank you for your insightful suggestion. Based on your feedback, we have included a detailed comparison with other comparative algorithms in the abstract to more comprehensively showcase the performance of our model. Specifically, we have added comparative experimental results between MAR-YOLOv9 and the FCOS model, demonstrating the significant improvement of MAR-YOLOv9 in terms of accuracy. These additional details not only enhance the completeness of the abstract but also provide readers with a clearer perspective to understand the innovative aspects and practical application value of our model.

2. The Introduction has explained the background of this paper and the purpose of the algorithm. The content of lines 168-174 in the Model construction part seems a bit lengthy. It is recommended to simplify it.

Response: Thank you for your meticulous review and valuable suggestions regarding our paper. In response to this feedback, we have made appropriate simplifications and modifications to the Model Construction section. We have strived to maintain the critical information while reducing redundancy and length. Now, this section more concisely and clearly introduces the structure and features of our proposed MAR-YOLOv9 model, making the article more reader-friendly and understandable.

3. For some reason, the pictures Fig1-Fig6 in the article seem to be compressed, not clear enough, and color deviation occurs.

Response: Thank you for your feedback. Regarding the issues with image clarity and color distortion, we have identified the cause to be an incorrect use of color space during the image processing, which led to color distortion and a decrease in image clarity. In response to this issue, we have made the necessary corrections. We have now reprocessed Figures 1 through 6 and the newly added Figure 7 to ensure accurate color representation and meet the clarity standards. Thank you for your patient guidance!

4. In the Backbone and Neck parts, there are no references and simple descriptions for modules such as Silence, SPPELAN, and CBFuse. It is recommended that the author add.

Response: Thank you for your valuable suggestions. We have carefully considered your requirements regarding the citation and description of the Silent, SPPELAN, and CBFuse modules within the paper. In response to your recommendations, we have added citations and corresponding brief descriptions of these modules in the Neck section of our model introduction. These additional contents will assist readers in better understanding the structure and working principles of the model, and further comprehend its functions.

5. In the Activation Functions part, the author lists three different activation functions and gives a brief description, but finally chooses one of them to build his own network. Therefore, the content of this section is a bit redundant, please consider deleting it as appropriate.

Response: Thank you for your suggestion. We have refined the section on the activation function, focusing on explaining why the SiLU activation function is the optimal choice for the MAR-YOLOv9 model and the object detection tasks in the agricultural domain. We have removed the discussion of other activation functions to allow readers to more directly understand our methodology and experimental results. We hope that this part of the article will now be more concise and focused.

6. It is recommended that the English expression of the paper be further optimized and improved to facilitate readers' understanding.

Response: Thank you for your suggestion. In order to enhance the quality of the English expression and ensure the clarity of the content in our paper, we have conducted a thorough language review and optimization of the entire text. We have paid special attention to the accuracy of grammar, consistency of terminology, and fluency of sentences to provide a better reading experience for the readers. We greatly appreciate your valuable comments and look forward to your further evaluation of the revised manuscript.

Finally, we appreciate very much for your time in editing our manuscript and the referees for their valuable suggestions and comments. I am looking forward to hearing from your final decision when it is made.

Sincerely,

Yangxu Wang

College of Robotics, Guangdong Polytechnic of Science and Technology, Zhuhai, China

E-mail: wangyx6432@gmail.com

---

## [Decision Letter · Decision Letter 1]

10 Jul 2024

MAR-YOLOv9: A Multi-Dataset Object Detection Method for Agricultural Fields Based on YOLOv9

PONE-D-24-14684R1

Dear Dr. Wang,

We’re pleased to inform you that your manuscript has been judged scientifically suitable for publication and will be formally accepted for publication once it meets all outstanding technical requirements.

Kind regards,

Xiaoyong Sun

Academic Editor

PLOS ONE

sunx1@sdau.edu.cn

Additional Editor Comments (optional):

Reviewers' comments:

Reviewer's Responses to Questions

**Comments to the Author**

1. If the authors have adequately addressed your comments raised in a previous round of review and you feel that this manuscript is now acceptable for publication, you may indicate that here to bypass the “Comments to the Author” section, enter your conflict of interest statement in the “Confidential to Editor” section, and submit your "Accept" recommendation.

Reviewer #2: All comments have been addressed

Reviewer #3: All comments have been addressed

2. Is the manuscript technically sound, and do the data support the conclusions?

Reviewer #2: Yes

Reviewer #3: Yes

3. Has the statistical analysis been performed appropriately and rigorously? 

Reviewer #2: Yes

Reviewer #3: Yes

4. Have the authors made all data underlying the findings in their manuscript fully available?

Reviewer #2: Yes

Reviewer #3: Yes

5. Is the manuscript presented in an intelligible fashion and written in standard English?

Reviewer #2: Yes

Reviewer #3: Yes

6. Review Comments to the Author

Reviewer #2: Thanks to the author for the careful and meticulous reply. All my doubts have been resolved. It is recommended that the paper be accepted.

Reviewer #3: accept～～～～～～～～～～～～～～～～～～～～～～～～～～～～～～～～～～～～～～～～～～～～～～～～～～～～～～～～～～～～～～～～～～～～～～～～～～～～～～～～～～～～～～～～～～～～～～～～～～～～～～～～～～～～～～～～～～～～～～～～～～～～～～～accept～～～～～～～～～～～～～～～～～～～～～～～～～～～～～～～～～～～～～～～～～～～～～～～～～～～～～～～～～～～～～～～～～～～～～～～～～～～

7. PLOS authors have the option to publish the peer review history of their article (what does this mean?). If published, this will include your full peer review and any attached files.

Reviewer #2: No

Reviewer #3: No

---

## [Editor Report · Acceptance letter]

17 Oct 2024

PONE-D-24-14684R1 

PLOS ONE

Dear Dr. Wang, 

I'm pleased to inform you that your manuscript has been deemed suitable for publication in PLOS ONE. Congratulations! Your manuscript is now being handed over to our production team.

Kind regards, 

on behalf of

Dr. Xiaoyong Sun 

Academic Editor

PLOS ONE